# Tunable quantum emitters on large-scale foundry silicon photonics

Hugo Larocque ◉[1] ✉, Mustafa Atabey Buyukkaya[2], Carlos Errando-Herranz[1,3], Camille Papon[1], Samuel Harper[2], Max Tao[1], Jacques Carolan[1,7], Chang-Min Lee[2], Christopher J. K. Richardson ◉[4], Gerald L. Leake[5], Daniel J. Coleman[5], Michael L. Fanto[6], Edo Waks[2] & Dirk Englund ◉[1]

Controlling large-scale many-body quantum systems at the level of single photons and single atomic systems is a central goal in quantum information science and technology. Intensive research and development has propelled foundry-based silicon-on-insulator photonic integrated circuits to a leading platform for large-scale optical control with individual mode programmability. However, integrating atomic quantum systems with single-emitter tunability remains an open challenge. Here, we overcome this barrier through the hybrid integration of multiple InAs/InP microchiplets containing high-brightness infrared semiconductor quantum dot single photon emitters into advanced silicon-on-insulator photonic integrated circuits fabricated in a 300 mm foundry process. With this platform, we achieve single-photon emission via resonance fluorescence and scalable emission wavelength tunability. The combined control of photonic and quantum systems opens the door to programmable quantum information processors manufactured in leading semiconductor foundries.

Coupling sources of quantum light to optical systems is a key requirement for several quantum photonic technologies ranging from quantum computations[1] to networking[2]. Among single photon emitters (SPEs), III–V semiconductor quantum dots (QDs) stand out for near-unity internal quantum efficiency, purity, and indistinguishability[3–9], making them key building blocks in technologies requiring on-demand entangled photon pair emission[10–12], photon–photon interactions[13–15], or photonic cluster state generation[16–19]. Recent advances in materials science and electron-nuclear spin control have also renewed interest in storing quantum information within these structures. Specifically, methods for reducing coupling between the QD electron spin with the nuclear spin bath of the embedding III–V material can push spin coherence times from tens of nanoseconds[20] to beyond 0.1 ms[21–23].

The central challenge now lies in developing systems for controlling many-QD quantum systems. This requires (i) efficiently

mediated optical interactions enabled by low propagation losses and (ii) scalable control of individual QDs. Addressing (i) and (ii) simultaneously has motivated the development of photonic integrated circuit (PIC) platforms as on-chip solutions for this quantum information processing task. Successfully deploying such integrated quantum technologies to their full potential critically hinges on the compatibility of QD structures with a given PIC platform. Leading approaches include monolithic III–V PICs[24,25], which have enabled interactions between multiple emitters[26–28]. However, optical attenuation in III–V waveguides exceeding 15 dB/cm[29] and the need for specialized manufacturing present obstacles on (i) and (ii), respectively. To address this constraint, recent work has pursued hybrid integration of quantum emitter materials with PIC platforms[30,31], including silicon[32–34] and silicon nitride[35–40]. However, none of these approaches combine (i) and (ii) with the scaling advantages of advanced foundry-based

[1]Research Laboratory of Electronics, Massachusetts Institute of Technology, Cambridge, MA 02139, USA. [2]Department of Electrical and Computer Engineering and Institute for Research in Electronics and Applied Physics, University of Maryland, College Park, MD 20742, USA. [3]Institute of Physics, University of Münster, 48149 Münster, Germany. [4]Laboratory for Physical Sciences, University of Maryland, College Park, MD 20740, USA. [5]State University of New York Polytechnic Institute, Albany, NY 12203, USA. [6]Air Force Research Laboratory, Information Directorate, Rome, NY 13441, USA. [7]Present address: Wolfson Institute for Biomedical Research, University College London, London, UK. ✉e-mail: hlarocqu@mit.edu

silicon-on-insulator (SOI) PIC platforms, which leverage general-purpose programmability, low power optical modulation, and the large-scale integration of thousands of individually controlled optical components[41–44]. Here, we address this challenge by simultaneously realizing four key advances: (a) iterative design, manufacture, and post-processing methods of advanced SOI PICs with sub-3 dB fiber coupling efficiency fabricated in a leading 300 mm foundry; (b) the hybrid integration of multiple telecom quantum emitters via scalable transfer printing methods; (c) resonance fluorescence from individually addressable SPEs; and (d) individual SPE emission wavelength tuning with the PIC's electronics.

## Results

### Architecture

Figure 1a illustrates our hybrid architecture, where we integrate the foundry SOI PIC with InP waveguide chiplets containing a central layer of InAs/InP QDs. As shown in the inset, the segmented top and bottom electrodes enable the application of electric fields primarily perpendicular to the QD layer to tune exciton emission via the quantum confined Stark effect[45,46]. We realize these electrodes with segmented bottom electrodes in p-doped silicon provided by the foundry process and a top-deposited ground electrode. In this configuration, local charge accumulations, modeled here as surface charges of $\pm\sigma$, lead to a floating gate effect introducing different levels of electric field screening along the InP waveguide.

We use confocal excitation to pump the QDs and modify the spatial distribution of surrounding carriers. QD emission into an InP waveguide chiplet thereafter couples via adiabatically tapered inverse couplers to the SOI PIC's waveguide layer for on-chip routing

or edge-coupling into silica fibers for off-chip routing. Figure 1b plots a representative photoluminescence (PL) spectrum collected through the edge-couplers at a base temperature of 5 K and a pump laser wavelength $\lambda_\mathrm{p}$ = 780 nm at 0.5 μW focused to a 0.92 μm spot-size. The spectrum shows QD emission in multiple, spectrally distinct lines in the telecom O-band; their density is consistent with ~10 QDs μm⁻².

We fabricated the PICs in the AIM Photonics' foundry. Using 193 nm deep-ultraviolet water-immersion lithography, this fabrication process enables 300 mm wafer-scale production of SOI PICs with multiple metal and dielectric layers along with available p- and n-type doping. The PIC used an O-band specific process development kit element for broadband optical edge couplers featuring loss below 3 dB over a 1260–1360 nm wavelength range[47] and other components custom designed for O-band operation. Facet-to-facet transmission through a straight 2 mm long silicon waveguide indicates a total transmission of 25 ± 5% while using 5 μm mode field diameter lensed fibers, indicating fiber-to-waveguide facet transmission better than 50% enabled by the multiple dielectric layers in this SOI PIC platform. Using ultra-high numerical aperture (NA) fibers with a mode field diameter of 4 μm can address the tighter optical confinement of our O-band photonic devices and potentially increase coupling efficiency to the 67% figure of merit specified by the edge coupler's design[47]. Figure 1c shows a wafer before dicing. Figure 1d highlights the PIC in this study, which occupies a 2 × 5 mm² block of the reticle.

### Hybrid integration

As outlined in the "Methods" section and in Supplementary Note 1, we develop a back-end-of-the-line process suited for a university

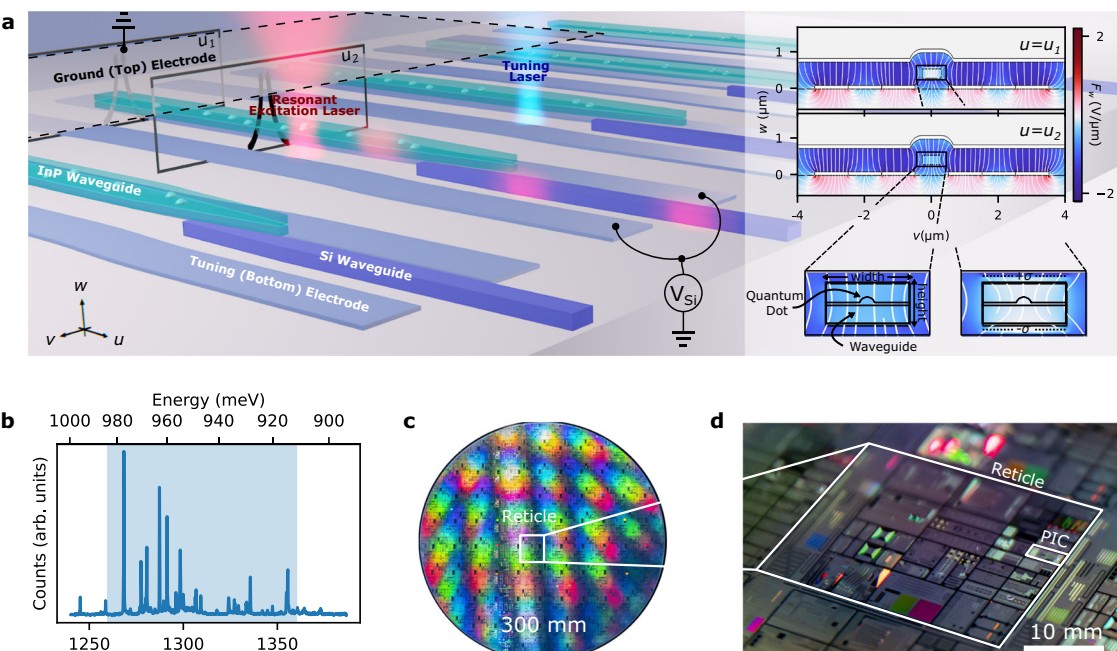

**Fig. 1 | Hybrid integration architecture. a** Schematics for foundry-based quantum photonics based on hybrid integrated tunable single-photon quantum emitters. Alignment between the waveguides of an InP chiplet and the silicon photonic integrated circuit (PIC) enables single photon adiabatic coupling in the resulting hybrid structure. We resonantly excite the emitters by focusing a laser beam onto the chiplet's emitters. Applying a voltage, $V_\mathrm{Si}$ between tuning and a ground electrode applies an electric field, represented as black lines, across the InP waveguide, thereby tuning the emission frequency of the enclosed QDs through the quantum confined Stark effect. Inset: finite element simulations of the electric field, white lines overlaid on the component $F_\mathrm{w}$ normal to the PIC surface, inside the InP

waveguide. QDs can experience various degrees of screening due to stray charges modeled here as surface charge densities of $\pm\sigma$. An additional laser beam with an energy above the bandgap of the chiplet can further affect these charge distributions, thus providing another mechanism for tuning the emission frequency of the QDs. **b** Representative photoluminescence spectrum from an ensemble of QDs, where the blue shading denotes the O-band, collected from photons that have been routed through the PIC. **c** Photograph of a 300 mm wafer on which our PIC was fabricated using a foundry process. **d** Closer view of a reticle from the wafer in **c** showing the location of our chip.

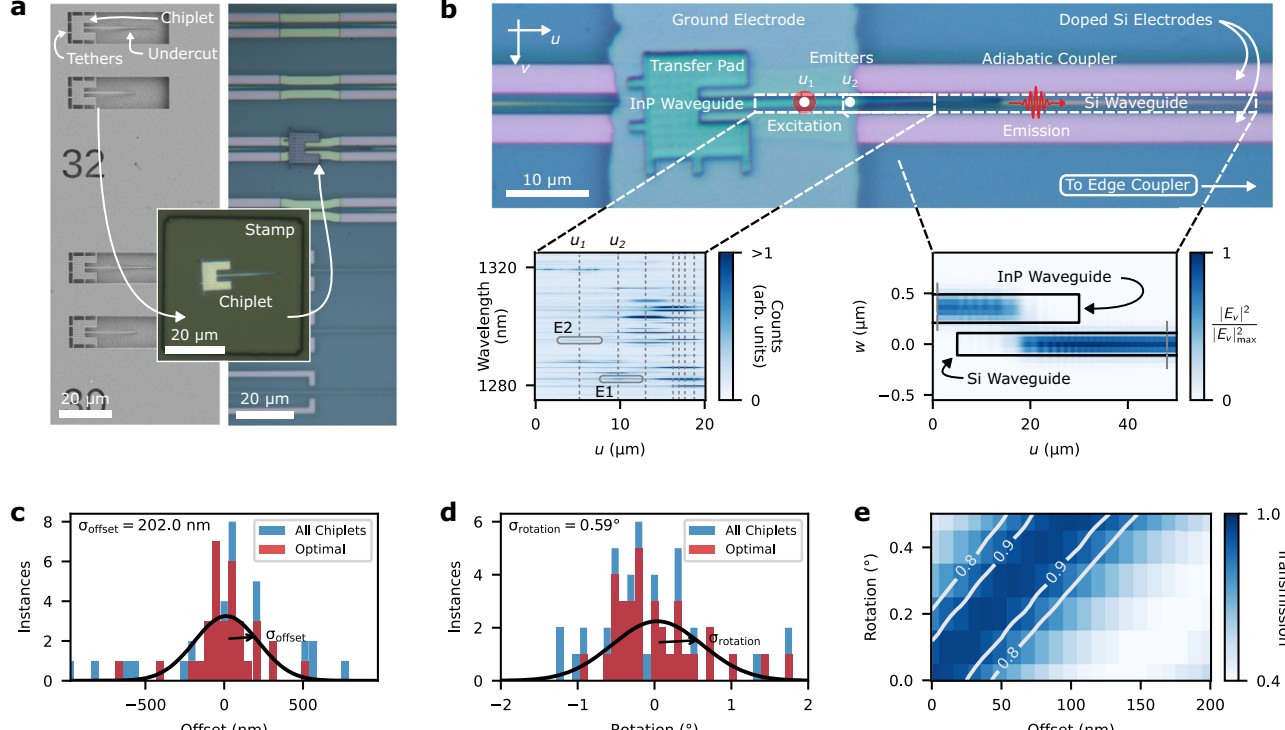

**Fig. 2 | Hybrid PIC Assembly. a** Scanning electron (left) and optical (right) micrographs of the suspended InP chiplets and of the hybrid photonic integrated circuit (PIC) at the transfer printing stage of its assembly, respectively. **b** Optical micrograph of the transferred microchiplet surrounded by the components tuning its emitters and routing single photons to the silicon-on-insulator (SOI) PIC. Left inset: photoluminescence spectra acquired while exciting emitters at various positions along the chiplet's nanobeam. Right Inset: Finite difference time domain simulation of the InP waveguide quasi-TE mode coupling from the chiplet to the PIC, where $E_v$ denotes the out-of-plane component of the electric field. **c** Relative offset and **d** rotation of 48 InP chiplets with respect to their underlying silicon waveguides on the PIC. The All Chiplets data set corresponds to data extracted from tapers attached to all the chiplets that were transferred onto SOI PICs. The Optimal data set only considers chiplets that did not experience any complications in their transfers due to residue underneath the chiplets. **e** Simulated optical transmission through the InP–silicon junction for various degrees of misalignment between the two tapers.

cleanroom dedicated towards integrating quantum devices on the foundry PIC. Figure 2a provides images of the chip at the key transfer printing step of the assembly process, where a 50 μm × 50 μm elastomer stamp picks and places an InP microchiplet with embedded InAs QDs[48] onto the SOI PIC. Optical monitoring through the stamp with a 0.41 NA objective ensures the alignment between the InP and silicon tapers.

Figure 2b shows an integrated chiplet at the end of the fabrication process, where we label the key components of the hybrid chip presented in the rendering from Fig. 1a. Here, the ground electrode covers the section of the nanobeam before the chiplet's adiabatic taper. Our experiments target emitters between the ground and doped silicon electrodes as they strongly overlap with the electric field of this capacitive structure. After wirebonding to a cryo-compatible printed circuit board, we mount the PIC in our cryostat and excite the QDs. We provide schematics of this experimental apparatus in Supplementary Note 2. The left inset in Fig. 2b presents a density plot of the QDs' PL spectra under the same conditions as Fig. 1b, acquired while sweeping the excitation laser spot along the waveguide axis $u$ while collecting out of the PIC facet into a single-mode fiber. This plot shows a number of distinct QD emission lines over a spatial extent of ~1.5 μm, which is consistent with the excitation laser spot size. The core results of this work focus on emitters located at $u_1$ and $u_2$, which we hereafter refer to as E1 and E2, respectively. The measured spectra feature stronger counts in the tapered part of the chiplet. As discussed in Supplementary Note 3, we attribute this observation to inhomogeneities in the transverse position of the QDs along the InP chiplet, relative to the center of the waveguide.

The facets of the silicon PIC and the InP–silicon taper junction account for most of our hybrid chip's optical losses. Uncertainties related to this metric primarily arise from those in our transfer printing process. For a perfectly aligned chiplet, finite difference time domain (FDTD) simulations indicate that the photon transfer efficiency from the InP's quasi-TE mode into the silicon waveguide's quasi-TE mode can reach values of up to 97.5%. The right inset of Fig. 2b provides a cross-section of such simulations after reaching convergence. To estimate the achievability of this metric, we measure transmission through a 2 mm silicon waveguide with a gap bridged by a double-sided InP taper. As further elaborated in Supplementary Note 4, we measured an adiabatic transfer of up to 86 ± 1% between the silicon and InP waveguides, which is also affected by structural differences between the chiplet used for this test and the ones shown in Fig. 2. Directly exciting the chiplet's quantum dots and measuring the emission counts from photons going through the silicon waveguides provides another path towards experimentally measuring losses caused by misalignment. However, as shown in Fig. 2b, differences in emission counts between QDs caused by their potentially varying quantum efficiencies and different locations in the InP waveguide add further uncertainty to this approach.

To establish a statistically achievable conversion efficiency for our transfer-printing method, we extract the misalignment of 48 adiabatic taper junctions across 6 fabrication iterations of hybrid chips. Figure 2c, d provides a histogram of the extracted offset and rotation, as defined in Supplementary Fig. 10, of the InP tapers with respect to the underlying silicon waveguides, respectively. We provide further details regarding the extraction of these parameters in Supplementary

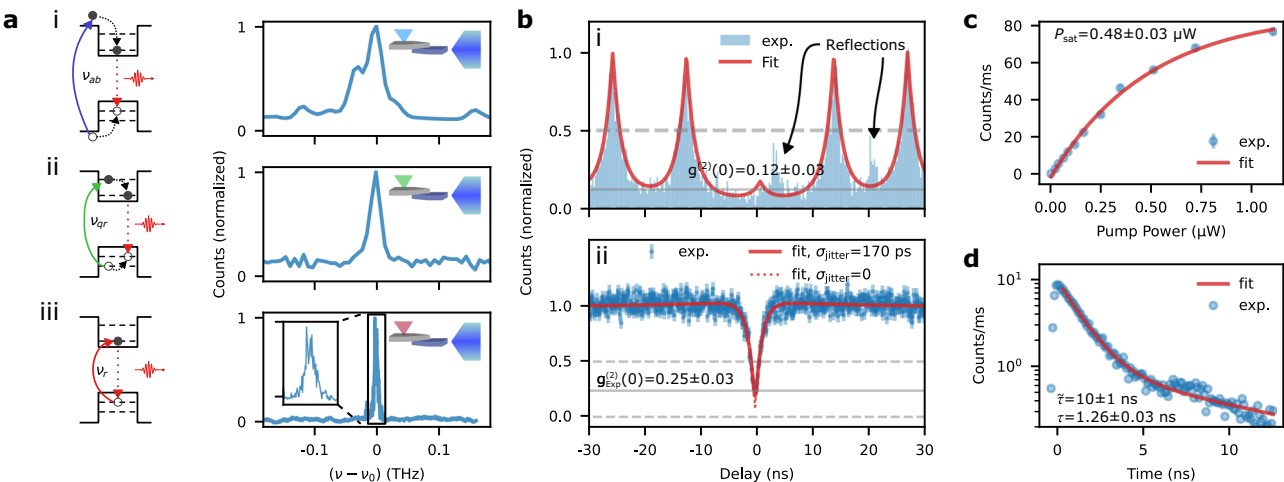

**Fig. 3 | Single-photon emission on a foundry chip. a** Emission spectra of emitter 1 (E1) under (i) above-band, (ii) quasi-resonant, and (iii) resonant excitation. All spectra are horizontally shifted by the emission frequency of the emitter, $\nu_0$. **b** Autocorrelation measurement, $g^{(2)}(\tau)$, under (i) triggered and (ii) continuous-wave excitation. Error bars correspond to the Poisson noise, $\sqrt{N}$, attributed to the detected counts, $N$. We provide fits of $g^{(2)}(\tau)$ that both incorporate and ignore detector jitter, $\sigma_{\text{jitter}}$. **c** Saturation curve of the emitter under pulsed above-band excitation indicating a saturation power $P_{\text{sat}} = 0.48 \pm 0.03\,\mu\text{W}$. **d** Radiative decay time trace of the emitter under pulsed above-band excitation indicating an emitter lifetime of $\tau = 1.26 \pm 0.03$ ns and a non-radiative lifetime of $\tilde{\tau} = 10 \pm 1$ ns.

Note 5. Under these optimal transfer printing conditions, the standard deviation of the data sets from Fig. 2c, d is given by 202 nm and 0.59°. With FDTD simulations similar to the one shown in Fig. 2b, we calculate the expected transmission through such imperfect junctions over this range of misalignment parameters, which we provide in Fig. 2e. We observe an average transmission of 69% over the statistically relevant range extracted from the data in Fig. 2c, d.

## Single-photon emission on a foundry chip

To demonstrate the versatility and performance of our system, we performed low-temperature photon correlation measurements on the PIC output facet under the three laser excitation regimes indicated in Fig. 3a: (i) above-band pumping, (ii) quasi-resonant excitation of a high-order excited state, and (iii) resonant excitation. The PL spectrum narrows to a linewidth of $15.5 \pm 0.6$ GHz when going from above-band to quasi-resonant excitation; this is near the 10 GHz resolution limit of the spectrometer that we use to measure (i) and (ii). Figure 3a(iii) shows the photoluminescence excitation (PLE) obtained by scanning an external cavity diode 200 kHz linewidth laser across the emitter at a laser intensity of ~10 μW/μm² while measuring the PIC output with superconducting nanowire single-photon detectors (SNSPDs). As seen from the inset, we estimate a linewidth of $5.8 \pm 0.2$ GHz. As emphasized in Supplementary Note 6, we found it unnecessary to use off-chip filtering under any excitation conditions during emission spectrum measurements, including the resonant case.

Next, we examine the purity of the excited photons with second-order autocorrelation measurements using an off-chip Hanbury Brown–Twiss apparatus consisting of a 50:50 fiber splitter and SNSPDs. We first triggered above-band excitation with a pulsed 776 nm fiber-coupled laser with an 80 MHz repetition rate and a 120 fs pulse width. Figure 3b provides the resulting autocorrelation trace of the emitted photons after going through a 0.1 nm narrow-band tunable fiber filter that we use to remove residual counts from nearby emitters also excited by above-band pumping. In spite of spurious peaks resulting from imperfections in the exciting pulses (see Supplementary Note 7), a fit of the data reveals clear anti-bunching at $g^{(2)}(0) = 0.12 \pm 0.03$. Figure 3b also provides the result of similar measurements using a continuous wave tunable O-band laser

resonantly exciting the emitter. We experimentally measure a $g^{(2)}_{\text{Exp}}(0) = 0.25 \pm 0.03$ from the resonance fluorescence. This measurement did not require any filtering due to good pump rejection inherited from the perpendicular geometry of our excitation and collection optics[49,50]. Our normalized raw $g^{(2)}_{\text{Exp}}(0)$ is similar to the value reported in other works that focus on collecting filter-free resonance fluorescence from near-infrared QDs coupled to integrated waveguides[50]. To estimate our photon purity, we fit our autocorrelation data to the response of a three-level model convoluted with the 170 ps response of our detectors. We provide the resulting fits overlaid with our experimental data along with the corresponding trace for an infinitely fast detector, which suggests a $g^{(2)}_{\text{Fit}}(0) = 0.08 \pm 0.03$. We provide additional fits to the autocorrelation trace in Supplementary Note 8 in order to further gauge effects related to dephasing, spectral diffusion, and blinking to other charge states or to the dark state. We also discuss how background counts from our pump likely result in our large $g^{(2)}_{\text{Fit}}(0)$. Additions to our measurement apparatus for improved photon purity through better pump rejection include polarization filtering[6], temporal gating[51], or bi-chromatic pumping[52,53].

Fitting the emitted photon count rate against the pump power to $R(P) = R_{\text{sat}}(1 - \exp(-P/P_{\text{E1,sat}}))$ indicates a saturation power of $P_{\text{E1,sat}} = 0.32\,\mu\text{W}$. These measurements are carried out with above-band excitation using an objective lens with an NA of 0.55, hence a saturation intensity of $I_{\text{E1,sat}} = 0.82\,\mu\text{W}/\mu\text{m}^2$. We find that this power value varies from 0.14 to 0.57 μW based on measurements for five emitters (see Supplementary Note 9). Figure 3c plots the saturation data and the corresponding fit for an emitter with a saturation power of $P_{\text{sat}} = 0.49\,\mu\text{W}$. As outlined in Supplementary Note 10, these measurements also allow us to estimate the efficiency of our device. By comparing our monitored counts to our laser's repetition rate and accounting for optical losses between our lensed fiber and detectors, we estimate a count rate of $1.04 \times 10^6$ counts/s at the fiber's output.

Figure 3d plots the time-resolved PL under pulsed above-band excitation, as measured on the SNSPDs. Fitting to a double-exponential decay $R(t) = a\exp(-t/\tau) + \tilde{a}\exp(-t/\tilde{\tau})$, where $a > \tilde{a}$, gives $\tau = 1.26 \pm 0.03$ ns and $\tilde{\tau} = 10 \pm 1$ ns, thereby suggesting a lifetime-limited linewidth of $\Delta\nu = 1/(2\pi\tau) = 126$ MHz. We attribute $\tau$ and $\tilde{\tau}$ to bright excitons within well-defined QDs and adverse electron-hole recombination processes,

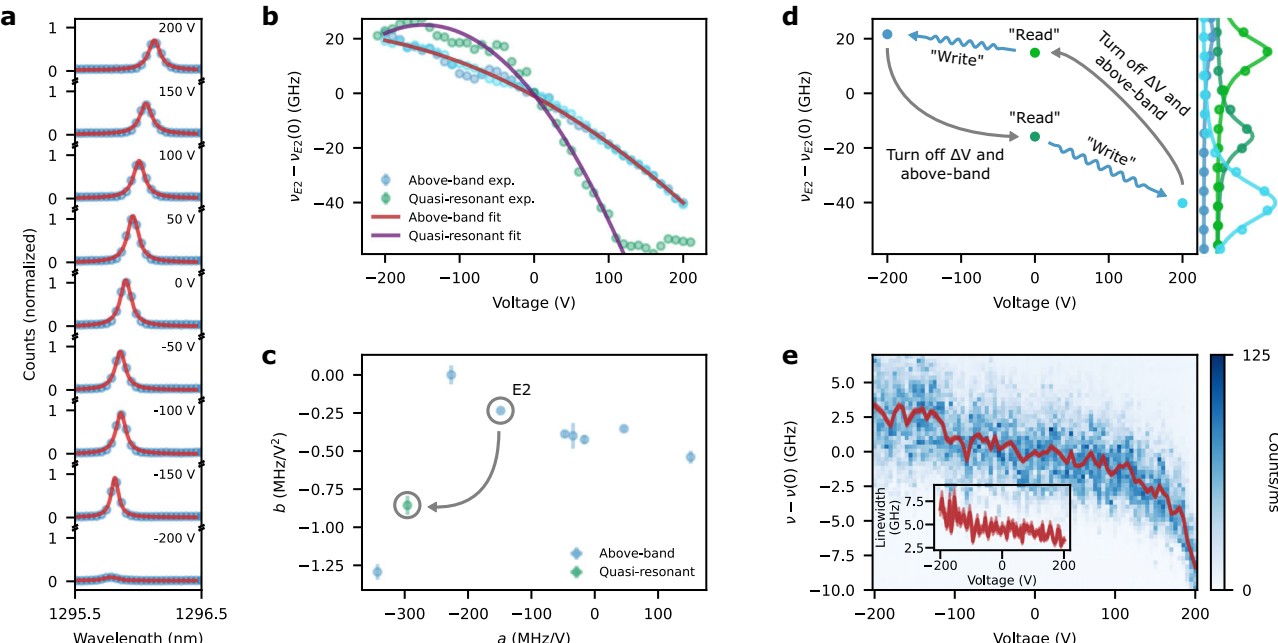

**Fig. 4 | Frequency tuning of individual quantum dots. a**, Emission spectrum and corresponding Lorentzian fits of emitter 2 (E2) under above-band illumination while voltages within ± 200 V are applied to the device. **b** Center wavelength of the Lorentzian fits shown in **a** in addition to data for other applied voltages, while the emitter is quasi-resonantly excited. **c** Best quadratic fit parameters, $a$ and $b$, to the shift experienced by eight different emitters extracted by fitting data of the type shown in (**b**). Error bars correspond to one standard deviation of the fitted parameter (see the "Methods" section). **d** Emission frequency of the quantum dot, $\nu_{E2}$, relative to its emission frequency at zero bias with above-band excitation, $\nu_{E2}(0)$, while writing it with above-band illumination at a bias and later reading them under quasi-resonant excitation with zero bias. The unbiased quasi-resonant spectra feature emission frequency shifts arising from the write step. **e** Photoluminescence excitation spectra of the resonance fluorescence from a biased quantum dot, where the red line indicates the average emission frequency of the spectra, $\nu$, relative to the dot's emission frequency at zero bias, $\nu(0)$. Inset: Fitted linewidth of the spectra.

respectively. Radiative decay traces for emitters with lower $\tau$ more prominently feature the influence of $\tilde{\tau}$ (see Supplementary Note 9).

## QD spectral tuning

We quantify the tunability of our QDs by examining the PL spectra of emitter E2 under above-band excitation while increasing the voltage applied to the device. Figure 4a presents E2's PL spectrum under continuous-wave above-band excitation as a function of the electric potential difference $\Delta V = V_{Si} - V_{gnd}$ between the tuning electrodes and the ground plane. We observe a voltage-dependent shift in the emission line's center frequency $\nu_{E2}(\Delta V)$ at an approximately constant PL intensity from $\Delta V = -150$ to 200 V. The PL linewidth and brightness are approximately constant within roughly $|\Delta V| < 150$ V. $\nu(\Delta V)$ is also consistent up to the spectrometer resolution across multiple measurements, thus suggesting that our tuning mechanism does not induce any significant noise (see Supplementary Note 11). We attribute the drop in counts at large negative biases to carrier tunneling. The conditions where tunneling effects become prominent likely depend on local material variations and charging, as the emission features of each dot differently respond to large bias voltages (see Supplementary Note 12).

Figure 4b plots the detuning $\nu_{E2}(\Delta V) - \nu_{E2}(0)$ obtained from Lorentzian fits to the spectra from Fig. 4a. Fitting these data to a quadratic model $\nu_{E2}(\Delta V) = \nu_{0,E2} + a_{E2}\Delta V + b_{E2}\Delta V^2$ gives a good agreement ($r^2 = 0.997$) for $a_{E2} = -148 \pm 1$ MHz/V and $b_{E2} = -0.23 \pm 0.01$ MHz/V$^2$. Figure 4c provides the fitted $a$ and $b$ for E2 and seven other emitters in the transferred chiplets. We observe an average $a = 0 \pm 100$ MHz/V, whereas the $b$ parameter falls around $b = -0.3 \pm 0.2$ MHz/V$^2$, indicating similar voltage sensitivities for our emitters. The quadratic frequency shift with a perpendicular electric field agrees with the form of the quantum confined Stark effect. As discussed in the Supplementary, stray

charges can lead to electric-field screening and commensurately reduce emission frequency shifts. The shifts shown in Fig. 4b, c readily highlight how additional free-carriers excited with the above-band laser constrain tuning, as quantified by $b_{ab} = -0.23 \pm 0.01$ MHz/V$^2$, compared to quasi-resonant excitation, where $b_{qr} = -0.85 \pm 0.06$ MHz/V$^2$. Though removing these actively generated carriers alleviates screening effects, previous reports[54–56] suggest the presence of other mechanisms attenuating $\Delta\nu(\Delta V)$ in our system.

To determine whether stray charges are further affecting emission, we measure the quasi-resonant spectrum of E2 at $\Delta V = 0$ after exposing the chiplet to above-band light while biased at $\Delta V \neq 0$. As further discussed in Supplementary Note 13, this type of biased illumination can write the underlying charge distribution around the quantum dot, and hence the screening conditions affecting emission read out with quasi-resonant excitation. Figure 4d plots how this write-read sequence affects the dot's emission frequency. Under unbiased quasi-resonant excitation, $\nu_{E2}$ falls within a range of 30 GHz, which we attribute to non-volatile charge redistribution near the QD. In Supplementary Note 13, we provide further discussions on how to leverage this writing procedure as a nonvolatile mechanism for individually tuning multiple QDs sharing the same tuning electrode.

To measure the influence of these stray charges under a minimally invasive setting, we measure $\nu(\Delta V)$ for a QD under resonant excitation, which does not introduce the additional charges and phonons that come with non-resonant excitation[57]. Figure 4e shows a density map of the PLE spectra from a biased quantum dot excited with 28 μW of optical power. We conducted this test on another dot given that E2 did not exhibit resonance fluorescence. As in the case of non-resonant excitation, the parabola formed by the center of the emission spectrum is asymmetric with respect to $\Delta V = 0$, which we attribute to an offset in the tuning electric field due to screening. The acquired

spectra also feature a significant degree of linewidth broadening, which we attribute to blinking due to charge noise. Both observations are consistent with stray charges affecting emission. We also notice that large forward biases mitigate the observed linewidth broadening, which could suggest the removal of residual charges around the QDs.

## Discussion

Future work should consider the following approaches to increasing the spectral tuning range of our device, narrowing the emitter linewidth, and leveraging other capabilities of advanced SOI PIC platforms. Unlike previous studies relying on the QDs investigated in this work[32,48,56], we were able to resonantly excite our emitters. As discussed in Supplementary Note 8, we consistently observe broad emission linewidths across five different QDs. Additional spectroscopy work on isolated QD samples could inform how our hybrid integration flow affects single-photon emission. Such measurements would lead to an improved post-processing flow able to narrow the emission linewidths of the QDs, leading to charge reductions and a path towards widely tunable and lifetime-limited on-chip SPEs at telecom wavelengths. Additionally, relying on charge stabilization with p–i–n junctions[4,27,58,59] or new telecom QD fabrication methods[2,60–64] could provide a viable path towards implementing such a platform. Such precautions would also assist in increasing our dots' indistinguishability[65], which was previously measured to be 20% and was primarily limited by nonradiative decay and charge noise[48].

The large-scale integration capabilities of silicon photonics make it ideally suited for hosting greater amounts of SPEs. Increasing the number of integrated chiplets would require a scalable transfer method, drawing on existing transfer printing techniques[66] used for large-scale hybrid integration of other light sources such as colloidal QDs and lasers[67,68]. If faced with electrical wiring limitations, further engineering the charging dynamics observed in Fig. 4d could provide a localized, non-volatile, and large-scale tuning mechanism for $\mathcal{O}(N)$ QDs with $\mathcal{O}(1)$ electrodes. Preventing displaced charges from directly contacting the III–V chiplets would avoid altering the coherence or tuning sensitivities of the QDs. Devices with large numbers of emitters could lead towards more sophisticated on-chip many-photon quantum systems that may also benefit from active components enabled by the platform such as RF electronics or cryogenics-compatible optical modulators[69]. Integrating other components such as on-chip SNSPDs[70,71] could also enable more complex types of on-chip photonic computations, including more complex quantum information processing tasks[72] and machine learning[73,74].

A combination of QD positioning methods used during chiplet fabrication[75,76] with large-scale schemes for optically probing nanophotonic structures[77] could efficiently prepare large numbers of QD chiplets for transfer. In fine-tuned industry settings, such precautions can lead to transfer rates of ~1 billion chiplets per hour[78]. For our hybrid chips, reaching such rates likely involves increasing the resolution of the optics monitoring the transfer, and adopting coupler designs that are more resilient to misalignment for the III–V silicon junction[79]. Preliminary screening of optical properties of emitters and improving their tunability beyond 100 GHz could lead towards on-chip sources spanning telecommunication windows.

In summary, we introduced a platform for integrating tunable single photon emitters on large-scale PICs. By hybrid integration of InAs/InP chiplets with a state-of-the-art SOI PIC, we have demonstrated scalable emission wavelength programmability with individual emitter resolution. The excellent pump rejection of the PIC enables the observation of QD resonance fluorescence without additional filtering. The localized non-volatile spectral tuning opens the door to QD wavelength tuning at the scale of the number of resolvable laser spots, which can be in the millions with conventional microscopes. However, the underlying spectral instability of the integrated quantum emitters constrains their usage for on-chip

quantum photonic applications. Overcoming this limitation will likely rely on emerging quantum dot growth methods[60–64] and state-of-the-art charge stabilization[27,59]. The combined control of photonic and atomic systems on a scalable platform enables a new generation of programmable quantum information processors manufactured in leading semiconductor foundries that can leverage many-body quantum systems.

## Methods

### Hybrid integration and post-foundry fabrication flow

We provide images of the fabrication flow for our hybrid PICs in Supplementary Fig. 1. Starting from the foundry-provided $2 \times 5\,mm^2$ PIC, where oxide and metal layers still cover the entire chip, we begin with the fabrication of 'quantum sockets' from the PIC surface to the silicon waveguides. After optical lithography and wet etching, we leave a 100 nm oxide layer over the waveguides to ensure good mechanical adhesion and optical coupling between the chiplet and the SOI PIC. This distance is fixed by the position of an etch stop layer used in the fabrication of the quantum sockets. This etch stop lies 100 nm over the silicon layer of the PIC, as determined by the layer stack used by the foundry. As shown in Supplementary Fig. 2, thinner oxide layers lead to better optical coupling between the InP and silicon waveguides. However, a smooth surface free of the topography formed by the patterned silicon waveguides and doped electrodes optimizes the chiplet transfer process outlined in Fig. 2a. In order to ensure a consistent oxide layer thickness from one hybrid integration run to the next, we choose to stick with the default thickness of 100 nm as specified by the foundry.

In parallel, we fabricate suspended InP chiplets with embedded InAs/InP QDs[48] by electron beam lithography and a combination of dry and wet etching. We then proceed with transfer printing chiplets from the parent InP chip into the PIC quantum sockets. This pick-and-place procedure uses an elastomer microstamp ($50\,\mu m \times 50\,\mu m$) tracked under a microscope. Finally, we deposit a 475 nm-thick PECVD oxide spacer layer and pattern a 20 nm Cr top electrode onto the quantum socket[56].

### Curve fitting

We use a nonlinear least-squares method to fit raw data to functions. The error on the various fitting parameters reported in this work corresponds to the square root of the respective diagonal element of the covariance matrix for the fitted optimal parameters.

## Data availability

The data that support the plots within this paper and other findings of this study are available from the corresponding author upon request and are also openly available at https://doi.org/10.6084/m9.figshare.26078704.

## Code availability

Code used to process the raw data into the results within this paper is available from the corresponding author upon request and is openly available at https://doi.org/10.6084/m9.figshare.26078704.

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

## Acknowledgements

H.L. and D.E. acknowledge support from the NSF (1936314QII-TAQS) and AFOSR program FA9550-16-1-0391. H.L. acknowledges the support of the Natural Sciences and Engineering Research Council of Canada (NSERC), the National Science Foundation (NSF, Award No. ECCS-1933556), Lawrence Berkeley National Laboratory (Research Subcontract no. 7571809 Prime Award: DE-AC02-05CH11231), and of the QISE-NET program of the NSF (NSF award DMR-1747426). C.E-H. acknowledges funding from the European Union's Horizon 2020 research and innovation program under the Marie Sklodowska-Curie grant agreement No.896401. M.A.B., S.H, C.-M.L., and E.W. acknowledge financial support from the National Science Foundation (grants #OMA1936314, #OMA2120757, #PHYS1915375, and #ECCS1933546), AFOSR (grant #FA23862014072), and the Maryland-ARL Quantum Partnership (W911NF1920181). C. P. acknowledges support of the NSF Convergence Accelerator program (Award No. 2040695) and of the NSF Engineering Research Center for Quantum Networks (Cooperative Agreement No. 1941583). The authors acknowledge James M. Daley for assisting with the sample's fabrication along with Marco Colangelo and Alessandro Buzzi for useful discussions. The authors acknowledge MIT.nano's facilities where additional fabrication work was performed. The simulations were performed using RSoft Photonic Device Tools distributed through Synopsys' University Software Program. The authors acknowledge X-Celeprint for distributing the elastomer stamps used in the assembly of the hybrid chip.

## Author contributions

H.L., J.C., and M.L.F. designed the silicon photonic integrated circuit. G.L.L., D.J.C., and M.L.F. oversaw its fabrication. C.J.K.R. grew the InAs/InP quantum dot sample. M.A.B., S.H., and C.-M.L. designed and fabricated the InP chiplets. H.L., M.T., and C.E.-H. transferred the chiplets on the silicon circuit. H.L. performed the back-end-of-the-line processing of the hybrid circuit. H.L., C.E.-H., and C.P. performed the experiment and analyzed the data. H.L. and M.T. performed FDTD simulations. D.E., E.W., and M.L.F. supervised all aspects of the project. All the authors contributed and discussed the content of this manuscript.

## Competing interests

The authors declare no competing interests.
