## [Peer Review File · Nature Communications]

REVIEWERS' COMMENTS

Reviewer #2 (Remarks to the Author):

I appreciate the detailed response to my comments and also to the comments of the other referees. The authors have addressed my previous questions and comments in a satisfactory way.

The manuscript has been clearly improved and the readability is much better than before. They also have now a very detailed supplementary section, explaining most of the relevant procedures, analysis and technical methods.

They showed an important and relevant technological step forward by implementing III-V semiconductor chipllets onto a 300 mm Si-wafer. In addition, they presented an extended characterization of such a system and explained the relevant physics. This is a significant step towards scalability of quantum photonic systems and a precondition for future applications in this field. However, I do not think that the achieved results in the quantum photonic properties are state-of-the art (quantum dot emission linewidth, wavelength tunability (stray fields), $g(2)$ -value etc.). The photons are far from being indistinguishable and therefore most of the intended applications in the field of on-chip quantum photonics (boson sampling, optical quantum computing, teleportation, entanglement swapping etc.) will not be possible.

Overall, I think the manuscript is now suitable for publication, but the limited quality of the quantum photonic properties will further limit the usability of such results for a multidisciplinary audience targeted by Nature Communications.

Reviewer #3 (Remarks to the Author):

The authors have satisfactorily attended to all of my previous comments, and have made significant alterations to their manuscript and supplementary information. I stand by my original suggestion that this work is of significant impact and should be published in Nature Communications. I have only 2 comments on the drafted documents below:

1. The paragraph describing the saturation is quite confusing (lines 236 to 249), switching between power and intensity throughout. As there is both the emitted photon flux (designated I) and also the pump intensity (also later using an I) I recommend using a different symbol for photon flux rather than I , and reserving I for the pump intensity. You could use R (for rate) or N (for number of photons) for example.

2. What is τ for the lifetime plots in the main text and supplement? These could be provided and would be of interest to readers.

Reviewer #4 (Remarks to the Author):

I commend the authors for seriously addressing my comments and those of my fellow referees. The revised manuscript is much improved. I still have a few concerns, but if these are addressed, I would recommend publication in Nature Comms.

1. My first concern relates to the answer of the authors to point 30 (in their response), which deals with the efficiency of the quantum dots as a function of their position in the taper. The authors explain this in terms of the orientation of the QD dipole relative to the waveguide/taper. I think that this is wrong because: 1. as far as I was aware, the QD dipoles were fixed by the crystal orientation so should be the same throughout the taper; 2. there are two orthogonal dipoles, so that even if one dies out, the second should kick in (e.g., Supp. Fig. 17b only considers one dipole). Maybe this has to do with where the QD is positioned relative to the (transverse) center of the waveguide? In any case, unless they can explain why I am wrong, they should fix their manuscript.

2. The use of Arbitrary Units throughout this work. I think that these might be used in places where they don't make sense, otherwise off quantities have been normalized that shouldn't be (and the results are either misleading or just wrong). Some examples I found with a quick scan: Supp. Table 1 - What does it mean to have Transmission in a.u.? Supp. Fig. 17b - what does an a.u. beta mean? Supp. Fig. 8d - Again Transmission in a.u., from which they extract an absolute efficiency somehow. Fig. 2e, main text - again Trans. in a.u. I stopped looking after these but wouldn't be surprised if there were other issues like this. This kind of thing (in conjunction with all of the errors the first time around) makes me kind of worried about whether there are other mistakes or misunderstandings throughout this work.

3. Sort of like 2: There are no units on the axes of Fig. 2e. I wouldn't accept this from undergraduate students and not in a Nature Comms paper. In any case, I assume that the rotation is in degrees, but couldn't guess what the units are on the Offset. I guess microns, but I would also note that here I think that offset refers to a vertical (v) misalignment in the frame of the figure, whereas later on (for sure in the supplementary) they use offset to discuss a distance along the waveguide/taper, which just makes the whole thing confusing.

We would like to thank the reviewers for their second round of comments. Provided below is a point-by-point reply forming the final revisions of our work. Passages that we explicitly added in the main manuscript to address these comments are colored in blue. Additions in the Supplementary were not annotated in order to conform with the journal's guidelines on final revisions.

We also restructured the Supplementary Information based on the order in which the Supplementary Notes are now mentioned in the main text. While referring to any new labeling of the Supplementary items, we refer to their original number and provide the reordered labels in parentheses.

1. **Reviewer #2 (Remarks to the Author):**

I appreciate the detailed response to my comments and also to the comments of the other referees. The authors have addressed my previous questions and comments in a satisfactory way.

The manuscript has been clearly improved and the readability is much better than before. They also have now a very detailed supplementary section, explaining most of the relevant procedures, analysis and technical methods.

They showed an important and relevant technological step forward by implementing III-V semiconductor chipllets onto a 300 mm Si-wafer. In addition, they presented an extended characterization of such a system and explained the relevant physics. This is a significant step towards scalability of quantum photonic systems and a precondition for future applications in this field. However, I do not think that the achieved results in the quantum photonic properties are state-of-the art (quantum dot emission linewidth, wavelength tunability (stray fields), $g(2)$ -value etc.). The photons are far from being indistinguishable and therefore most of the intended applications in the field of on-chip quantum photonics (boson sampling, optical quantum computing, teleportation, entanglement swapping etc.) will not be possible.

Overall, I think the manuscript is now suitable for publication, but the limited quality of the quantum photonic properties will further limit the usability of such results for a multidisciplinary audience targeted by Nature Communications.

We thank the Reviewer for their assessment of our work. As mentioned in our first reply, we agree that the spectral properties of our integrated quantum dot emitters limit the scope in which our hybrid platform can perform various tasks pertaining to on-chip quantum photonics. We have added the following passage to the last paragraph of the main text to explicitly acknowledge this limitation:

However, the underlying spectral instability of the integrated quantum emitters constrain their usage for on-chip quantum photonic applications. Overcoming this limitation will likely rely on emerging quantum dot growth methods [60-64] and state-of-the-art charge stabilization [27,59].

2. Reviewer #3 (Remarks to the Author):

The authors have satisfactorily attended to all of my previous comments, and have made significant alterations to their manuscript and supplementary information. I stand by my original suggestion that this work is of significant impact and should be published in Nature Communications. I have only 2 comments on the drafted documents below:

We thank the reviewer for their assessment of our work. We hope that our second round of edits satisfactorily addresses their comments.

1. The paragraph describing the saturation is quite confusing (lines 236 to 249), switching between power and intensity throughout. As there is both the emitted photon flux (designated I) and also the pump intensity (also later using an I) I recommend using a different symbol for photon flux rather than I , and reserving I for the pump intensity. You could use R (for rate) or N (for number of photons) for example.

We thank the reviewer for pointing out the confusion that comes from our interchangeable notation. We agree that using count rates and the symbol R (for rate) to account for emitted photons clarifies the presentation of our results. Based on this feedback, we have reworded the first part of the paragraph describing saturation as follows:

Fitting the emitted photon count rate against the pump power to $R(P)=R_{\text{sat}}(1-\exp(-P/P_{E1,\text{sat}}))$ indicates a saturation power of $P_{E1,\text{sat}}=0.32 \mu\text{W}$.

We correspondingly modified other passages that rely on such count rate measurements, such as determining the emitter's lifetime:

Fitting to a double-exponential decay $R(t) = a \exp(-t/\tau) + \tilde{a} \exp(-t/\tilde{\tau})$, where $a > \tilde{a}$, gives $\tau = 1.26 \pm 0.03 \text{ ns}$ and $\tilde{\tau} = 10 \pm 1 \text{ ns}$, thereby suggesting a lifetime-limited linewidth of $\Delta\nu = 1/(2\pi\tau)=126 \text{ MHz}$.

These notational changes have also been brought in the captions of Supplementary Figures 15 & 16 (now Supplementary Figures 17 & 18).

2. What is $\tilde{\tau}$ for the lifetime plots in the main text and supplement? These could be provided and would be of interest to readers.

We agree with the reviewer that the $\tilde{\tau}$ might be of interest to the readers and have thus included their values in the main text and the supplementary.

Fitting to a double-exponential decay $R(t) = a \exp(-t/\tau) + \tilde{a} \exp(-t/\tilde{\tau})$, where $a > \tilde{a}$, gives $\tau = 1.26 \pm 0.03$ ns and $\tilde{\tau} = 10 \pm 1$ ns, thereby suggesting a lifetime-limited linewidth of $\Delta\nu = 1/(2\pi\tau) = 126$ MHz.

We have also included the fitted $\tilde{\tau}$ for the radiative decay traces in Supplementary Figure 16 (now Supplementary Figure 18).

5. Reviewer #4 (Remarks to the Author):

I commend the authors for seriously addressing my comments and those of my fellow referees. The revised manuscript is much improved. I still have a few concerns, but if these are addressed, I would recommend publication in Nature Comms.

We thank the Reviewer for acknowledging our efforts addressing their first round of comments and hope that our revisions to the second round are equally satisfactory.

1. My first concern relates to the answer of the authors to point 30 (in their response), which deals with the efficiency of the quantum dots as a function of their position in the taper. The authors explain this in terms of the orientation of the QD dipole relative to the waveguide/taper. I think that this is wrong because: 1. as far as I was aware, the QD dipoles were fixed by the crystal orientation so should be the same throughout the taper; 2. there are two orthogonal dipoles, so that even if one dies out, the second should kick in (e.g., Supp. Fig. 17b only considers one dipole). Maybe this has to do with where the QD is positioned relative to the (transverse) center of the waveguide? In any case, unless they can explain why I am wrong, they should fix their manuscript.

We thank the reviewer for pointing this out. The main text and the supplementary now attributes the disparity on the coupling efficiency to the transverse offset of the QD position across the waveguide:

As discussed in Supplementary Note 3, we attribute this observation to inhomogeneities in the transverse position of the QDs along the InP chiplet, relative to the center of the waveguide.

However, as shown in Fig. 2b, differences in emission counts between QDs caused by their potentially varying quantum efficiencies and different locations in the InP waveguide adds further uncertainty to this approach.

Supplementary Addition: The results from Supplementary Figure 6 suggest that variations in the monitored counts amongst emitters likely stem from variations in

the transverse position of the quantum dots along the InP chiplet relative to the center of the waveguide

We added the corresponding β factor for such offsets in Supplementary Figure 17a (now Supplementary Figure 6a) for both the X and Y dipoles, as labeled by newly added dashed lines. As the reviewer pointed out, the orientation will be fixed for both dipoles after fabrication and any losses incurred in the Y dipole due to its imperfect orientation in the waveguide should be compensated by gains attributed to the X dipole, whose orientation differs by 90° relative to the Y dipole. The orientation dependence of Supplementary Figure 17a (now Supplementary Figure 6a) should illustrate how these two contributions balance out for different degrees of transverse offsets between the quantum dot's position and the center of the waveguide. However, the varying amplitudes of these curves also show that quantum dots positioned along the waveguide's center lead to better light-matter interactions.

We also included additional curves in Supplementary Figure 17b (now Supplementary Figure 6b) illustrating the coupling efficiency of the X dipole to the tapered waveguide mode. In light of our addition to Supplementary Figure 17a (now Supplementary Figure 6a), we also added the corresponding coupling efficiency curves for the Y and X dipoles for a quantum dot with a 70 nm transverse displacement relative to the center of the waveguide. The narrowest waveguide width of the taper considered in this plot is 177.5 nm, which is located at a 15 μm distance down the taper. Simulations considering a 70 nm displacement therefore nicely illustrate the effects of transverse displacement on optical coupling efficiency, as it ensures that the quantum dot still falls within the waveguide towards the end of the considered tapered region.

Given the more numerous parameters relating the position of the quantum dot relative to the geometry of the InP waveguide, we have provided an additional panel in Supplementary Figure 17d (now Supplementary Figure 6d) that clarifies the various "offsets" and distances considered in this figure.

7. 2. The use of Arbitrary Units throughout this work. I think that these might be used in places where they don't make sense, otherwise off quantities have been normalized that shouldn't be (and the results are either misleading or just wrong). Some examples I found with a quick scan: Supp. Table 1 - What does it mean to have Transmission in a.u.? Supp. Fig. 17b - what does an a.u. beta mean? Supp. Fig. 8d - Again Transmission in a.u., from which they extract an absolute efficiency somehow. Fig. 2e, main text - again Trans. in a.u. I stopped looking after these but wouldn't be surprised if there were other issues like this. This kind of thing (in conjunction with all of the errors the first time around) makes me kind of worried about whether there are other mistakes or misunderstandings throughout this work.

We have been using arbitrary units to represent quantities that are either unitless, normalized, or represent a fraction. On that note, the units in Supplementary Table 1

represent absolute units. The units in Supplementary Figure 8 and 17 (now Supplementary Figures 8 and 6) are also absolute. We apologize for the confusion that this caused. We removed all instances of “arbitrary units” in figures with transmission ratios:

- Figure 2e
- Supplementary Figure 2b (now Supplementary Figure 2b)
- Supplementary Figure 8c,d (now Supplementary Figure 8c,d)
- Supplementary Table 1
- Supplementary Figure 17a,b,c (now Supplementary Figure 6a,b,c)

Data that has been normalized now explicitly indicate this with our use of “normalized” as opposed to “arb. units” in the relevant labels:

- Figure 3 a,b
- Figure 4a
- Supplementary Figure 13 (now Supplementary Figure 16)
- Supplementary Figure 14 (now Supplementary Figure 13)
- Supplementary Figure 18 (now Supplementary Figure 7)
- Supplementary Figure 19 (now Supplementary Figure 19)
- Supplementary Figure 20 (now Supplementary Figure 20)
- Supplementary Figure 22d,e (now Supplementary Figure 22d,e)

8. 3. Sort of like 2: There are no units on the axes of Fig. 2e. I wouldn't accept this from undergraduate students and not in a Nature Comms paper. In any case, I assume that the rotation is in degrees, but couldn't guess what the units are on the Offset. I guess microns, but I would also note that here I think that offset refers to a vertical (v) misalignment in the frame of the figure, whereas later on (for sure in the supplementary) they use offset to discuss a distance along the waveguide/taper, which just makes the whole thing confusing.

We thank the Reviewer for raising this point. The units of Fig. 2e were expressed in terms of the standard deviation values of the fitted distributions shown in Figs. 2c,d and were thus unitless. To remove any further confusion, we changed the units of Fig. 2e to degrees and nanometers.

The use of the terminology “Offset” in Supplementary Figures 6,7 (now Supplementary Figures 10,11) refers to the same transverse offset from Fig. 2 of the main text. As pointed out by the reviewer, the term “Offset” in Supplementary Figure 17 (now Supplementary Figure 6) does indeed refer to the position along the taper. To remove any further confusion, we now refer to this offset as “Distance along InP waveguide” in the supplementary.